# The Role of *SMAD4* Inactivation in Epithelial–Mesenchymal Plasticity of Pancreatic Ductal Adenocarcinoma: The Missing Link?

**DOI:** 10.3390/cancers14040973

**Published:** 2022-02-15

**Authors:** Marie-Lucie Racu, Laetitia Lebrun, Andrea Alex Schiavo, Claude Van Campenhout, Sarah De Clercq, Lara Absil, Esmeralda Minguijon Perez, Calliope Maris, Christine Decaestecker, Isabelle Salmon, Nicky D’Haene

**Affiliations:** 1Department of Pathology, Erasme University Hospital, Université Libre de Bruxelles (ULB), 1070 Brussels, Belgium; laetitia.lebrun@erasme.ulb.ac.be (L.L.); andrea.schiavo@erasme.ulb.ac.be (A.A.S.); claude.van.campenhout@erasme.ulb.ac.be (C.V.C.); sarah.de.clercq@erasme.ulb.ac.be (S.D.C.); lara.absil@erasme.ulb.ac.be (L.A.); esmeralda.minguijon.perez@erasme.ulb.ac.be (E.M.P.); calliope.maris@erasme.ulb.ac.be (C.M.); isabelle.salmon@erasme.ulb.ac.be (I.S.); nicky.d.haene@erasme.ulb.ac.be (N.D.); 2DIAPath, Center for Microscopy and Molecular Imaging (CMMI), Université Libre de Bruxelles (ULB), 6041 Gosselies, Belgium; christine.decaestecker@ulb.be; 3Laboratory of Image Synthesis and Analysis, Brussels School of Engineering/École Polytechnique de Brussels, Université Libre de Bruxelles (ULB), 1050 Brussels, Belgium

**Keywords:** pancreatic ductal adenocarcinoma, PDAC, epithelial–mesenchymal transition, EMT, epithelial–mesenchymal plasticity, SMAD4, biomarker, metastasis

## Abstract

**Simple Summary:**

Pancreatic ductal adenocarcinoma (PDAC) is currently one of the deadliest cancers. Despite the progress that has been made in the research of patient care and the understanding of pancreatic cancer, the survival rate remains mediocre. *SMAD4*, a tumor-suppressor gene, is specifically inactivated in 50–55% of pancreatic cancers. The role of SMAD4 protein loss in PDAC remains controversial, but seems to be associated with worse overall survival and metastasis. Here, we review the function of *SMAD4* inactivation in the context of a specific biological process called epithelial–mesenchymal transition, as it has been increasingly associated with tumor formation, metastasis and resistance to therapy. By improving our understanding of these molecular mechanisms, we hope to find new targets for therapy and improve the care of patients with PDAC.

**Abstract:**

Pancreatic ductal adenocarcinoma (PDAC) presents a five-year survival rate of 10% and its incidence increases over the years. It is, therefore, essential to improve our understanding of the molecular mechanisms that promote metastasis and chemoresistance in PDAC, which are the main causes of death in these patients. *SMAD4* is inactivated in 50% of PDACs and its loss has been associated with worse overall survival and metastasis, although some controversy still exists. SMAD4 is the central signal transducer of the transforming growth factor-beta (TGF-beta) pathway, which is notably known to play a role in epithelial–mesenchymal transition (EMT). EMT is a biological process where epithelial cells lose their characteristics to acquire a spindle-cell phenotype and increased motility. EMT has been increasingly studied due to its potential implication in metastasis and therapy resistance. Recently, it has been suggested that cells undergo EMT transition through intermediary states, which is referred to as epithelial–mesenchymal plasticity (EMP). The intermediary states are characterized by enhanced aggressiveness and more efficient metastasis. Therefore, this review aims to summarize and analyze the current knowledge on SMAD4 loss in patients with PDAC and to investigate its potential role in EMP in order to better understand its function in PDAC carcinogenesis.

## 1. Introduction

Pancreatic ductal adenocarcinoma (PDAC) is the fourth leading cause of cancer death worldwide, with a five-year-survival rate at 10%, which is the lowest among all cancer types [1]. The incidence of pancreatic cancer has continued to increase over the years, and it is predicted to become the second leading cause of mortality related to cancer in 2030 [1,2].

Its extremely poor prognosis is due to several different factors. Pancreatic cancer is usually diagnosed at an advanced stage due to a lack of symptoms as well as diagnostic and prognostic tumor markers [3]. Consequently, only 10–20% of patients present with resectable pancreatic cancer at diagnosis [3,4,5].

For patients with unresectable or borderline resectable PDAC, the most commonly employed therapies include neoadjuvant chemotherapy and/or radiotherapy. Likewise, surgery is also usually followed by adjuvant chemotherapy in an attempt to improve overall survival (OS). Unfortunately, survival remains mediocre due to the intrinsic resistance of pancreatic cancer to conventional therapies, including chemotherapy, radiotherapy and targeted molecular therapy [3,5].

Although some progress has been made in the research of patient care and the understanding of the molecular and genetic background of pancreatic cancer, the improvement of OS for PDAC in the past two decades has been modest [6,7]. The accession of modified FOLFIRINOX as an adjuvant therapy might have modestly increased disease-free survival (DFS) and OS, but at the expense of a greater number of toxic effects [8].

More effective therapeutic strategies are, therefore, desperately needed. These include the implementation of better screening strategies and the discovery of tumor markers allowing for earlier detection. The development of targeted therapies and the identification of subgroups of patients who could benefit from individualized therapies are also required. Improving our understanding of pancreatic cancer carcinogenesis, namely of the molecular mechanisms that promote metastatic spread and chemoresistance, in an attempt to optimize the current treatments of pancreatic cancer, is indispensable.

PDAC has been extensively studied throughout the years and its genomic aberrations have been well-described [9,10,11,12,13,14,15]. However, despite the increasing precision and depth of coverage, the four main driver genes identified before the time of next-generation sequencing (NGS) remain on the forefront: the oncogene *KRAS*, which is mutated in almost 95% of PDACs together with alterations of the tumor-suppressor genes *CDKN2A*, *TP53* and *SMAD4* [9,16,17]. These mutations are acquired in a sequenced time of events, with *KRAS* and *CDKN2A* being affected early in the carcinogenesis of PDAC, while *TP53* and *SMAD4* characterize later events, predominantly occurring in invasive PDAC [18,19].

*SMAD4* alterations are relatively specific to gastrointestinal cancers, notably pancreatic cancer, where *SMAD4* is inactivated in 50–55% of cases [20,21,22,23,24]. SMAD4 is a member of the Smad family of transcription factor proteins and is the central signal transducer of the transforming growth factor-beta (TGF-beta) signaling pathway. This signaling pathway is well known for its role in inducing epithelial–mesenchymal transition (EMT). 

EMT is the process through which cells with epithelial characteristics acquire a mesenchymal cell phenotype and behavior [25,26]. EMT has been increasingly studied in the last two decades for its role in carcinogenesis, metastasis and therapy resistance. However, recent research suggests that EMT is not a binary process, and that cells transition through a spectrum of intermediary states. Each state presents its own characteristics and behavior. The capacity of cells to shift through partial EMT states is referred to as epithelial–mesenchymal plasticity (EMP) [26].

The biological mechanisms underlying the role of SMAD4 in EMT and the metastatic process in PDAC remain unclear and are a subject of controversy. 

This review aims to summarize the current knowledge concerning the role of *SMAD4* alterations in patients with PDAC and to investigate its potential role in EMT, particularly in that of EMP, in order to better understand its function in terms of the aggressiveness of PDAC.

## 2. Literature Review

In order to investigate the published studies on the TGF-beta pathway, SMAD4 and EMT in patients with PDAC, a comprehensive literature search of the electronic database PubMed was performed up to January 2022. Studies were selected using the following search terms (non-exhaustive list): “pancreas”, “pancreatic cancer”, “ pancreatic ductal adenocarcinoma”, “PDAC”, “TGF-beta”, “SMAD4”, “epithelial-mesenchymal transition”, “EMT”, “hybrid EMT”, “partial EMT” and/or “epithelial-mesenchymal plasticity”.

## 3. Discovery and General Structure of SMAD4

The *SMAD4* gene is located on the 18q21.2 human chromosome locus. It is composed of 12 exons, of which the second to twelfth are translated, with the first one containing just part of the 5′ untranslated region (5′UTR), and coding for a 552 amino acid sequence resulting in a 60 kDa protein [21,23,24,27].

*SMAD4* was first identified as a tumor-suppressor gene by Hahn et al. in 1996. Its original name was initially designated as “Deleted in pancreatic cancer 4” (*DPC4,* homozygously deleted in pancreatic carcinoma, locus 4) [21]. Its current name, “Smad”, is derived from a combination of the names of the genes that code for two orthologous proteins: the “small” worm phenotype (sma) from *Caenorhabditis elegans* and Mothers against decapentaplegic (Mad) from *Drosophila melanogaster*. Indeed, the human protein sequence of SMAD4 presents similarities to the *D. melanogaster* Mad protein and to the *C. elegans* Mad homologs sma-2, sma-3, and sma-4, with SMAD4 being the human homolog of sma-4 [28,29].

The SMAD4 protein structure is composed of two functional sites: the N-terminal Mad homology 1 (MH1) domain, which essentially allows SMAD4 to bind to DNA, and the C-terminal Mad homology 2 (MH2) domain, which enables homo- and hetero-oligomerization with other Smad proteins. Both domains may interact with other proteins, but it is mainly the MH2 domain that can interact with transcription factors, co-activators and co-repressors. These two domains are connected by a linker region that serves predominately as a regulatory site [29,30].

## 4. SMAD4 and TGF-Beta Signaling Pathway

SMAD4 is the central mediator of the TGF-beta signaling pathway, which has key roles in the development and homeostasis of epithelial cells, stromal compartments and immune cells in the gastrointestinal system. However, the TGF-beta signaling pathway is also implicated in carcinogenesis, which is commonly called the “TGF-beta paradox” or the “TGF-beta switch” [31].

There are two branches in the TGF-beta family signaling pathway: the TGF-beta branch, represented by ligands such as TGF-beta, activin, nodal, or myostatin, and the bone morphogenetic protein (BMP) branch, represented by ligands such as BMP and growth and differentiation factor (GDF) [32,33]. In the present review, we will focus on the TGF-beta branch and the implication of SMAD4 in this pathway. 

There are three different TGF-beta ligands (TGF-beta 1–3), which are almost identical and play virtually the same roles in vitro [25]. Three different types of TGF-beta receptors (TGFBR 1 to 3) also exist. TGFBR1/2 enable intracellular signaling through their intrinsic enzymatic activity, while TGFBR3 acts as a regulator of the TGF-beta pathway [23,25,30,34]. On the surface of human cells, TGFBR1 and TGFBR2 exist essentially as homodimers in the absence of a ligand. The TGF-beta ligand binds specifically to TGFBR2, which leads to the formation of tetra-heteromeric receptor complexes between TGFBR1 and TGFBR2. This enables the phosphorylation of TGFBR1 by TGFBR2 [23,30,34]. TGFBR1 subsequently phosphorylates the intracytoplasmic receptor-regulated Smad proteins (R-Smad), SMAD2 and SMAD3. They then proceed to form a heteromeric complex with the only known common-Smad protein (Co-Smad), SMAD4. This protein complex translocates to the nucleus to regulate gene transcription in association with different transcription factors (TFs) and co-activators/co-repressors, in a gene- and cell-specific manner [23,25,30,34]. The Smad protein complexes are incapable of direct recruitment of the basal transcription machinery to responsive promoters. They instead regulate gene transcription through chromatin remodeling and histone modifications [30]. When the TGF-beta signaling pathway acts through SMAD2/3, it is referred to as the “canonical TGF-beta pathway” (Figure 1).

SMAD6 and 7 are inhibitory-Smads (I-Smads) that have a regulatory effect on the pathway. This regulation is possible by binding directly to R-Smads, preventing their recruitment or phosphorylation, but also by promoting the ubiquitin proteolysis of TGFBR1 and promoting its dephosphorylation [23,30,34].

The TGF-beta pathway can also function in the absence of SMAD2/3, which is called the “non-canonical TGF-beta pathway”. Besides Smad-mediated transcription, TGF-beta can activate the MAP kinases ERK1/2, JNK and p38MAPK; the cell survival mediators PI3K, AKT1/2 and mTOR; the inflammation mediators NF-kB, COX-2 and prostaglandins; small GTP-binding proteins such as Ras, RhoA, Rac1 and Cdc42; and the nonreceptor protein tyrosine kinases PP2A, Src, FAK and Abl [23,25,34]. Although the activation of these pathways are Smad independent, they can influence Smad-mediated pathways by regulating Smad activation, inducing or repressing Smad-mediated transcription. 

## 5. The Role of the TGF-Beta Signaling Pathway in PDAC: In Brief

In epithelial cells, the TGF-beta pathway plays a dual role, functioning both as a tumor suppressor and a tumor promoter. It is not certain how this switch is operated, but it has been suggested that the canonical and non-canonical TGF-beta pathways play a tumor-suppressive role at the early stages of tumor development in pancreatic cancer. Indeed, in the early stages of PDAC, high TGF-beta1 expression may counteract proliferation and is associated with better OS [35]. In contrast, during the stages of disease progression, it will promote invasion, migration and metastasis [23,25,30,34,36]. Patients with high TGF-beta expression and advanced disease present significantly reduced survival [37,38].

TGF-beta has potent cytostatic effects and is known to regulate the cell cycle by inducing cyclin-dependent-kinase (CDK) inhibitors, such as CDKN2B and/or CDKN2A, depending on the cell type [33,39].

In a physiological setting, the TGF-beta pathway also triggers apoptosis, although the exact biological mechanism is still undefined. The TGF-beta signaling pathway may induce the expression of death-associated protein kinase (DAP-kinase) in a Smad-dependent manner [40]. Increased expression of SMAD4 has also been linked to apoptosis [41,42].

TGF-beta can restrict epithelial cell proliferation and tumor formation by acting on the surrounding stroma as well as the inflammatory cells. Indeed, TGF-beta will, for example, limit the expression of mitogenic factors secreted by the nearby fibroblasts, inhibiting epithelial growth [33,39]. However, when the TGF-beta switch takes place, the microenvironment is modulated in a manner that promotes epithelial cell proliferation. The composition of the tumor microenvironment may also change, enabling invasion and migration and inducing chemoresistance [25,33,39,43]. TGF-beta is also a potent immunosuppressor and promotes immune tolerance by regulating the activities of macrophages, natural killer (NK) and effector T lymphocytes [33,39]. Therefore, in the context of carcinogenesis, TGF-beta may permit the immune evasion of cancerous cells. 

One of the most notable tumor-promoting mechanisms induced by the TGF-beta canonical and non-canonical pathways is EMT. The description of this process as well as its role in PDAC shall be thoroughly discussed later on in this review.

## 6. SMAD4 and Pancreatic Ductal Adenocarcinoma

In 2008, Jones et al. first introduced the concept of core signaling pathways based on high-throughput Sanger sequencing of 20,661 protein-coding genes. Their analysis demonstrated that the TGF-beta signaling pathway was affected in 100% of PDACs, with *SMAD4* being the most frequently altered gene [9]. *SMAD4* is mainly inactivated by two mechanisms, predominantly by homozygous deletion or mutations with the loss of the wild-type allele [21,44]. These somatic alterations include nonsense, missense and frameshift mutations, usually leading to a truncated and/or a non-functional protein. More than 80% of these alterations occur on the MH2 functional domain, with a hotspot of missense mutations occurring on codon 361 [21,24,45]. MH2 domain mutations can cause a failure to homo- and hetero-oligomerize, while MH1 domain mutations prevent DNA binding by SMAD4. [24,45,46]. It has also been demonstrated that some mutations in both the MH1 and MH2 domain can increase degradation via ubiquitin-mediated proteolysis [46,47,48]. It is interesting to note that larger genetic aberrations affect the *SMAD4* gene; notably, a loss of heterozygosity (LOH) of the 18q (where *SMAD4* is located) occurs in 90% of PDACs [21].

Immunohistochemistry (IHC) is a reliable method to evaluate the genetic status of *SMAD4*, due to it being a sensitive and specific marker (Figure 2) [27]. Wilentz et al. detected no difference in immunolabeling between PDACs presenting with homozygous deletion or intragenic mutations associated with allele loss. Additionally, a single wild-type allele was sufficient to exhibit “diffusely positive staining” [27]. Therefore, SMAD4 detection by IHC can be used in routine pathology to detect *SMAD4* genetic alterations, which can be a useful marker in diagnosing PDAC on cytology, biopsy or surgery specimens [22,49,50].

In PDAC, a loss of SMAD4 expression has been associated with tumor size, peripancreatic extension, TNM staging, lymphatic invasion, nodal involvement and poor differentiation [51,52,53], influencing OS [54,55,56,57,58,59,60,61,62,63,64] and DFS [56,59,61,64].

However, the impact of SMAD4 loss remains controversial. Several large retrospective studies have not been able to statistically demonstrate a prognostic impact of *SMAD4* alterations on OS and DFS [10,52,65,66,67,68,69,70,71]. In some studies, *SMAD4* inactivation is a negative prognostic factor solely in a specific subset of patients. Hsieh et al. associated SMAD4 loss with worse DFS, but only in patients presenting with homozygous deletion and not *SMAD4* mutations [72]. 

Similarly, SMAD4 deficiency alone was not always linked to prognosis, but its loss, associated with other biomarkers, could predict OS and/or DFS to a significant statistical degree [37,73,74]. For example, in a study conducted by Yokose et al., *SMAD4* alterations alone could not predict OS or DFS, but the combination of *KRAS* and *SMAD4* mutations was an independent poor prognostic factor in PDAC [75]. In the same way, Park et al. did not find any statistical difference in terms of OS in patients with SMAD4 loss, but poorly differentiated tumors that were SMAD4-negative were better predictors of survival [51].

The implication of SMAD4 in therapy resistance is equally debated. It has been suggested that patients with SMAD4-negative tumors present drug resistance and that they do not benefit form adjuvant chemotherapy [62,76]. Indeed, a recent meta-analysis focusing on SMAD4-related drug resistance suggests that the loss of SMAD4 is associated with worse OS and DFS in patients receiving adjuvant chemotherapy [77]. In a series of patients with PDAC receiving downstaging therapy, patients with intact SMAD4 tumors had better treatment response and tumor regression scores [78]. Similarly, patients with SMAD4-deficient locally advanced PDAC receiving radiofrequency ablation presented worse DFS after treatment. [79]. Furthermore, in in vitro and in vivo assays, SMAD4-negative PDAC cells were associated with increased radioresistance [80].

On the contrary, in certain studies, patients with SMAD4-negative tumors seemed to benefit from adjuvant chemotherapy and present longer OS or DFS [65,66,67]. Biankin et al. even indicated that SMAD4 loss co-segregated with resectability and that patients had better survival after surgery [10].

Most notably, SMAD4-negative status appeared to be correlated with the tendency to metastasize rather than to recur locally [56,74,76,81,82,83,84]. Yachida et al. identified the genetic subtype of PDAC characterized by a *TP53* missense mutation and biallelic loss of *SMAD4* which was linked with higher metastatic efficiency [85]. In an experimental setting, using genetically engineered mouse models (GEMMs), Bardeesy et al. evidenced an association between *SMAD4* mutations and an increased number of liver metastases [86]. Similarly, functional SMAD4, by repressing specific genes such as *FOSL1*, suppresses metastatic colonization. If SMAD4 is lost, FOSL1 can promote migration and metastatic colonization [87]. Therefore, SMAD4 detection in PDAC could help in orienting patient treatment.

However, the propensity of SMAD4-negative tumors to metastasize widely is also controversial. In certain clinical studies, SMAD4 loss was not associated with a specific recurrence pattern. Indeed, SMAD4 deficiency did not correlate with locoregional nor distant recurrence [66,68,70]. 

In conclusion, further investigations are needed to understand the prognostic value of *SMAD4* alterations in PDAC.

## 7. An Introduction to Epithelial–Mesenchymal Transition

In a physiological context, EMT is indispensable for development and wound healing. However, EMT has been increasingly associated with human disease, particularly in the cellular mechanisms of invasion, migration, metastasis and microenvironment remodeling in cancer [25]. 

EMT is the process through which cells transition from an epithelial state to a mesenchymal one. [25,26]. The epithelial state is defined by epithelial characteristics such as apical-to-basal polarization and tight cell-to-cell junctions in addition to the expression of epithelial markers, such as E-cadherin or cytokeratins [25,26]. In the mesenchymal state, cells acquire a fibroblast-like appearance and upregulate the expression of mesenchymal cell markers, including N-cadherin and vimentin [25,26]. 

One of the most important signaling pathways regulating EMT is the TGF-beta pathway [25,26,33,88,89,90]. TGF-beta signaling is known to induce EMT in a Smad-dependent and -independent manner through the expression of specific EMT transcription factors (EMT-TFs) such as Snail, Slug, Twist and ZEB1 [23,25,33,90]. 

For years, EMT has been described as a binary process, where cells transition from one state to the other completely and irreversibly. Recently, it was suggested that cells can pass through a variety of intermediary states, and express a mixture of epithelial and mesenchymal features, in a reversible manner [26,90]. Each intermediate state is defined by specific markers and behavior [26,90,91,92]. This type of EMT, previously called “hybrid” or “partial” EMT, is now preferably referred to as “epithelial-mesenchymal plasticity” (EMP) as it better describes the ability of cells to adopt mixed features and shift readily through the epithelial–mesenchymal spectrum [26]. Studies have indicated that cells transitioning through these intermediate states are characterized by higher tumor-initiation properties and metastatic potential than those on either side of the EMT spectrum, due to the co-expression of mesenchymal and epithelial markers [92,93,94].

## 8. Epithelial–Mesenchymal Transition in PDAC: What Is Known

Numerous studies have focused on the role of EMT in PDAC in order to understand the underlying mechanism of metastasis and drug resistance, mostly in in vitro and in vivo settings. 

The detection of epithelial and mesenchymal markers using either IHC, immunofluorescence (IF) or RNA sequencing (RNA-Seq) has suggested the existence of EMT in PDAC [95,96,97,98,99,100,101,102]. The EMT signature appeared to be associated with a poor prognosis, having an impact on OS and DFS [96,97,98,100,101,102,103].

It has also been suggested that EMT may be an extremely early event in PDAC, playing an essential role in distant dissemination and patient prognosis. Indeed, in an experimental setting, the presence of EMT was detected in high-grade preneoplastic lesions and in circulating tumor cells (CTCs) before identifiable tumor formation [104,105]. 

However, the necessity of EMT in metastasis initiation is still a matter of debate. Using GEMM, studies have suggested that metastasis can occur in PDAC independently of EMT [106,107,108]. On the other hand, Krebs et al. indicated that the EMT-TF ZEB1 was indispensable for PDAC to metastasize; its knockout impacted PDAC tumorigenicity on multiple levels, notably that of cancer cell plasticity [109]. Additionally, the upregulation of EMT-TFs increased metastasis in vivo, suggesting a potential role of EMT in PDAC metastasis [110]. 

Another property of EMT is that this process could confer chemoresistance. In vitro studies demonstrated that chemoresistance of PDAC cells lines was associated with the expression of EMT markers such as ZEB1 [111,112]. Similarly, the loss of EMT-TFs Snail and Twist correlated with chemosensibility in vitro and in vivo [106].

To conclude, EMT is a complex cellular process that might influence the aggressiveness of PDAC. However, the exact underlying mechanisms are still not fully understood. We will, therefore, review the potential role of *SMAD4* alterations in order to improve our understanding of their impact on EMT in PDAC.

## 9. *SMAD4* Alterations and EMT in PDAC

Although SMAD4-dependent TGF-beta activation is associated with later stages of PDAC carcinogenesis and EMT, paradoxically, it is SMAD4 loss that appears to be linked to worse prognosis and metastasis. Therefore, it is not fully understood how TGF-beta-induced EMT operates in the absence of SMAD4.

Most studies suggest that EMT requires an intact TGF-beta signaling pathway, including SMAD4 [86,107,113,114,115,116,117,118]. For example, SMAD4-intact cell lines and tumors in GEMMs were capable of inducing EMT and acquiring a mesenchymal phenotype. On the other hand, SMAD4-deficient cell lines and GEMM tumors maintained a more epithelial identity when exposed to TGF-beta [86,117]. Similarly, EMT was identified in a subset of GEMMs with typical PDAC gene mutations (*KRAS* mutations + *TP53* or *CDKN2A* alterations) but intact TGF-beta signaling [119,120]. Moreover, using RNA-Seq analysis on PDAC specimens, Chan-Seng-Yue et al. demonstrated that “Classical-like A/B” signatures, known to be associated with better prognosis and less EMT, correlated with *SMAD4* inactivation. However, EMT markers were linked to PDACs presenting aggressive gene signatures, called “Basal-like A/B” [101]. 

Interestingly, in the early stages of carcinogenesis, it has been suggested that TGF-beta may have a tumor-suppressor effect by inducing a lethal EMT [113,118]. In SMAD4-intact cells, TGF-beta induced EMT and apoptosis via the EMT-TF Snail as well as the repression of KLF5 expression. In this situation, SOX4, known to be an important factor in tissue differentiation and present in progenitor pancreatic cells, will initiate the transcription of proapoptotic factors. In the absence of SMAD4, KFL5 is not repressed, and—conjointly with SOX4—will transcribe a protumorigenic gene program. This biological process might explain the recurring “TGF-beta switch” observed in PDAC and mirrors the empirical observations made in clinical studies.

In contrast, some studies indicate that tumors can exhibit EMT when *SMAD4* is inactivated [52,56,121]. Indeed, in a retrospective clinical study that classified PDAC specimens in epithelial, hybrid and mesenchymal phenotypes using an IHC score, SMAD4 loss was found to be correlated with the mesenchymal phenotype, worse outcome and distant recurrence [56]. Moreover, Wartenberg et al. observed an association between *SMAD4* inactivation and the “immune-escape” subtype, which was characterized by worse outcome and a high rate of EMT tumor budding [121]. Additionally, the *SMAD4* Y353C missense mutation was linked with increased migration and EMT markers in vitro [52].

On a different note, Levy et al. suggested, using *SMAD4* siRNA cell lines, that the loss of SMAD4 in cancer might contribute to cell cycle arrest and migration, but not EMT [122]. Indeed, it is well-known that some of the non-canonical TGF-beta pathways are capable of inducing EMT. Downstream kinases such as Ras, Src, PI3K, MAPK, Par6 and NF-kB have elicited Smad-independent EMT and have been extensively reviewed [123,124]. Therefore, it is possible that a certain redundancy in the TGF-beta pathway exists. If *SMAD4* is inactivated, other pathways may be stimulated and promote TGF-beta-associated EMT (Table 1).

In conclusion, the biological function of *SMAD4* alterations in PDAC EMT has not been completely elucidated.

## 10. Controversies Regarding *SMAD4* Alterations and its Potential Role in PDAC EMP

As mentioned above, the prognostic value of *SMAD4* alterations is still a subject of controversy and the exact biological function of SMAD4 in PDAC, specifically in the TGF-beta-induced EMT, is not yet fully understood.

Firstly, we think that the controversy around the clinicopathological and prognostic role of SMAD4 loss could be related to a number of different factors: (1) There is a lack of standardization in the histopathological study of SMAD4 IHC. Indeed, there is a high heterogeneity in the use of antibodies and the evaluation of SMAD4 IHC staining. This could explain the large difference in the proportion of SMAD4- negative tumors across studies. (2) Although SMAD4 IHC has been described as a reliable tool, discrepancies exist between protein expression and gene which do not always lead to a complete absence of staining [52,69,125]. Additionally, some studies have indicated that the nature of the genetic alterations has a significant impact on survival [72,125]. SMAD4 protein expression and genetic alterations may have to be analyzed simultaneously to determine specific PDAC phenotypes. (3) The discrepancies could also be linked to the patient selection regarding disease stage and/or treatment. The initial hypothesis that SMAD4 loss is related to an enhanced metastatic potential was evoked in an autopsy series by Iacobuzio-Donahue et al. in 2009, which mostly included patients with advanced disease [81]. Therefore, it is unclear if this hypothesis could be applied to patients who have undergone resection and received neoadjuvant and/or adjuvant therapy. The prognostic impact and pattern of recurrence of SMAD4 loss was also studied in patients with PDAC resection, but there was a great variety of adjuvant therapy protocols [10,53,58,60,61,65,66,74,76,82,83,125].

Secondly, diverging results were observed regarding the role of SMAD4 in PDAC EMT. One possible explanation is related to the lack of standardization and criteria to evaluate EMT, and more particularly EMP [26]. Indeed, the use of the term “EMT”, mainly in the study of cancer, has created a great number of discrepancies in data interpretation and disagreements as to whether the studied process is EMT or not [26]. Based on the Consensus Statement mediated by The EMT International Association (TEMTIA), there exist multiple issues concerning the characterization of EMT: (1) EMT cannot solely be defined by the expression of one or few biomarkers. EMT is characterized by the downregulation of epithelial markers to acquire a mesenchymal phenotype. However, diverse markers have been used to define EMT, which has been a major source of confusion. For example, in some studies in PDAC, EMT was defined only by the complete or partial loss of E-Cadherin and the upregulation of vimentin [56,100,117,126]. (2) There are no guidelines as how to define EMT based on the cellular and molecular markers. There exists a very important heterogeneity in the combination of the EMT-TFs and/or mesenchymal markers used to determine EMT and/or EMP states [95,96,97,99,101,102]. In addition, the up-regulation of EMT-TFs and other biomarkers may be context- and tissue-specific [110]. For example, the dysregulation of Wnt/beta-catenin plays an important role in EMT in colorectal carcinoma, but its role in pancreatic cancer is less clear and somewhat controversial [127]. Furthermore, the regulation of EMT also operates via the post-transcriptional regulation of EMT regulators at both the mRNA and protein levels [26,128,129]. (3) EMT is a complex process characterized by the modulation of the expression of diverse biomarkers, but also changes in cellular properties and characteristics. This is particularly true for studies that use RNA expression to exclusively survey EMT molecular markers [102]. Indeed, EMT is also defined as the loss of apical-to-basal polarity, tight cell junctions, the acquisition of spindle-cell-like characteristics and increased motility. These changes must also be analyzed and taken into account when defining EMT [25,26]. (4) It is equally important to note that EMT remains a dynamic process. A certain EMT phenotype could be a definitive state of the tumor or a transient state in time.

Finally, we hypothesize that the disagreements concerning the biological function of SMAD4 in PDAC might be due to its implication in the process of EMP. Indeed, the majority of research on *SMAD4* alterations and EMT is focused on the binary model of EMT, rather than considering the context of EMP. However, the existence of EMP in pancreatic cancer has been evoked in various studies [56,108,130,131,132,133,134]. Aiello et al. demonstrated that PDAC cells can disseminate according to two different EMT programs: a partial and a complete EMT program [130]. The complete EMT program is characterized by cells that are derived from poorly differentiated tumors, repress epithelial markers, enhance mesenchymal markers and migrate in a single-cell manner. On the other hand, the partial EMT program is characterized by cells that derived from well-differentiated PDACs, co-express epithelial as well and mesenchymal markers and migrate in clusters of CTCs. It has been suggested that cells migrating in clusters, although less motile, have enhanced seeding capacity [135,136]. This would suggest that tumor cells in partial EMT states present a higher metastatic potential than cells that undergo a complete EMT. Indeed, the increased aggressivity of intermediate EMT states has already been evoked and mentioned above [92,93]. Interestingly, in a study conducted by Huang et al. using PDAC organoids, *SMAD4* inactivation enabled a collective invasion program upon TGF-beta exposure, while organoids with wild-type *SMAD4* invaded with a mesenchymal phenotype [108]. Moreover, in a GEMM of squamous cell carcinoma, later stages of partial EMT, closer to a mesenchymal phenotype, presented upregulation of SMAD2/3 [92]. Taken together, it is tempting to suggest that cells presenting *SMAD4* alterations, even if SMAD2/3 are upregulated, acquire an early partial EMT state, closer to an epithelial phenotype, characterized by enhanced metastatic potential.

## 11. Conclusions

Despite the fact that progress has been made in the research of patient care and the molecular and genetic background of pancreatic cancer, the improvement of OS for PDAC is relatively modest and it remains one of the deadliest cancers of our time. It is, therefore, imperative to improve our understanding of pancreatic cancer carcinogenesis, namely of the molecular mechanisms that promote metastatic spread and chemoresistance, which are the main causes of death in pancreatic cancer.

It was previously demonstrated that the TGF-pathway is affected in all cases of PDACs, mainly by *SMAD4* alteration [9]. Knowing that the TGF-beta pathway plays a crucial role in initiating EMT, we decided to explore the implication of SMAD4 in EMT and, more particularly, in the EMP of PDAC. Indeed, empirical observations have suggested that SMAD4 loss in PDAC is correlated with more metastasis. 

However, it is important to note that SMAD4 signaling can be triggered by other impulses than TGF-beta. SMAD4 is also the central mediator of the BMP signaling pathway. While the TGF-β pathway is often disrupted in pancreatic cancer, alterations in the BMP pathway have not been frequently reported. The BMP pathway plays a role in tumorigenesis, but its role in pancreatic cancer remains poorly understood [86,137]. Concerning EMT, BMP proteins are capable of inducing EMT in pancreatic cell lines and are dependent on SMAD4 function [138,139]. However, the inactivation of BMP receptors and/or loss of SMAD4 have also been associated with increased proliferation, invasiveness and poor prognosis [140]. Seeing a similar controversy as that observed in the TGF-beta branch, it would therefore be interesting to study the different branches of the TGF-beta family and their potential mutual influence on EMT.

Although the biological function of *SMAD4* alterations in EMT are not completely elucidated, *SMAD4* inactivation may play an important role by inducing partial EMT states. These partial EMT states harbor both epithelial and mesenchymal traits and markers, and may have enhanced metastatic potential. 

A better understanding of the molecular mechanisms underlying PDAC, *SMAD4* and EMT could help in the development of effective targeted therapy, which is highly needed in PDAC [141]. 

## Figures and Tables

**Figure 1 cancers-14-00973-f001:**
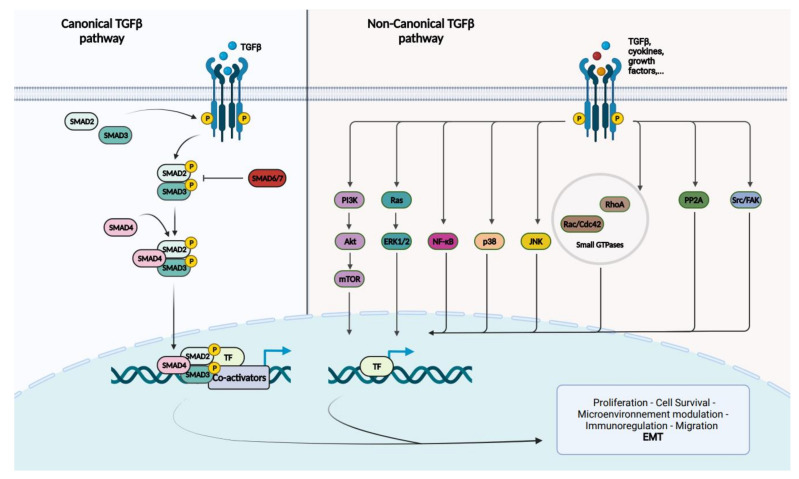
Schematic representation of the canonical and non-canonical TGF-beta signaling pathways. TGF-beta or other types of ligands will bind to TGFBR2, which recruits and phosphorylates TGFBR1. In the canonical TGF-beta pathway, TGFBR1 phosphorylates R-Smads, SMAD2 and SMAD3. The Co-Smad, SMAD4, will form a heteromeric complex with the phosphorylated R-Smads and translocate to the nucleus. Once in the nucleus, Smad proteins interact with transcription factors (TFs) and co-activators/co-repressors to regulate gene transcription. I-Smads, SMAD6 and SMAD7 regulate TGF-beta signaling. The non-canonical TGF-beta signaling regroups MAP kinases including ERK1/2, p38 and JNK; the cell survival mediators PI3k/Akt/mTOR; inflammation mediators, such as NF-kB; small GTP-binding proteins, such as Ras, RhoA, Rac and Cdc42; and nonreceptor protein tyrosine kinases including PP2A, Src and FAK.

**Figure 2 cancers-14-00973-f002:**
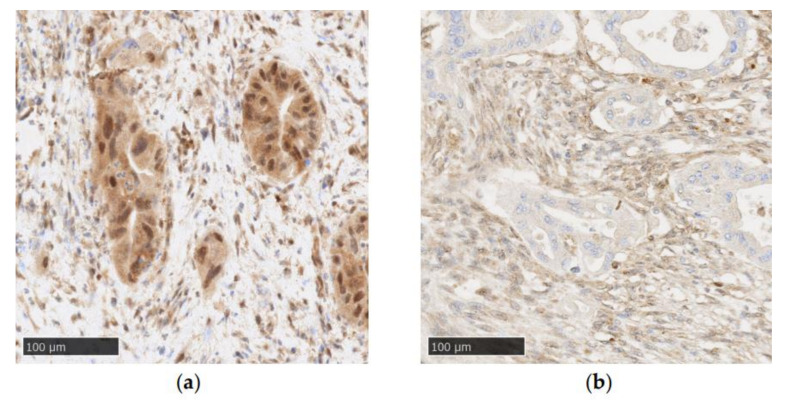
Examples of SMAD4 immunohistochemistry in PDAC (Recombinant Anti-SMAD4 antibody (EP618Y), Abcam, original magnification: ×20): (**a**) positive SMAD4 staining with tumor cells showing cytoplasmic and nuclear staining; (**b**) negative SMAD4 staining characterized by a loss of nuclear and cytoplasmic staining in tumor cells.

**Table 1 cancers-14-00973-t001:** Review of the literature: role of the SMAD4 protein in EMT.

Authors	Year	Material	Methods	Results
EMT requires intact SMAD4				
Bardeesy N et al. [86]	2006	PDAC GEMM	WB (E-CAD, SLUG and SMAD4) IHC (E-CAD, CK19 and SMAD4) IF (E-CAD, SLUG) Cell proliferation assay reagent, wound-healing assay	SMAD4-null tumors displayed significantly more prominent epithelial identity including higher E-CAD and lower SLUG expression, upholding a role of SMAD4 in promoting EMT.
Zhao S et al. [114]	2008	Isogenically matched PDAC cell lines (BxPC3, Capan-2, MIAPaCa-2, CFPAC-1, PANC-1 and UK PAN-1) Orthotopic mouse model	WB (Beta-CAT, E-CAD, ERK1/2, phosphorylated ERK1/2Thr202/Tyr204, SMAD2/3, SMAD4, phospho-Smad2, STAT3, STAT3Ser727, phospho-STAT3Tyr705 and VIM) IHC (phospho-STAT3Tyr705) shRNA (STAT3) Clonogenic assay, Matrigel invasion assay	Cells expressing SMAD4 showed an enhanced TGF-beta-mediated EMT as determined by the increased expression of VIM and decreased expression of Beta-CAT and E-CAD.
Chen YW et al. [115]	2014	PDAC cell lines (AsPC-1, CFPAC-1 and PANC-1) Mouse model	WB (Akt/p-AKT, CD133/44, CD133/1 c-Jun/p-c-Jun, Fast-1, Fos, E-CAD, EGFR/p-EGFR Hes1, nestin, NF-kB, p-p44/42, PTEN, SMAD2/3, SMAD4, Sp1, TGF-beta1, VIM) IHC (CD133, E-CAD, EGFR, nestin, SMAD4) IF (SMAD4) RT-qPCR (CD133, CD44, E-CAD, EGFR, N-CAD, SMAD4, VEGF and VIM) shRNA (SMAD4) Transient transfections, luciferase reporter assays, cell proliferation assay, wound-healing assay and Transwell migration assay	SMAD4 deficiency in vitro promotes an epithelial phenotype and induced chemoresistance. SMAD4 restoration in vivo increased migration and EMT markers (VIM and SMA).
Kang Y et al. [116]	2014	PDAC cell line (PANC-1) Human cell line (HPNE)	WB (phosphor-Akt, CK19, phosphor-MEK1/2, MEK1/2, N-CAD, p21, phospho-SMAD2, phosphor-SMAD3, SMAD2/3, SMAD4, Tak1, VIM and Wafl/Cip1) IHC (N-CAD, SMAD4) IF (CK19, SMAD4) RT-PCR, RT-qPCR (Beta-CAT, E-CAD, FN1, N-CAD, TWIST1, TWIST2, VIM and ZEB1) ChIP (N-CAD, SMAD4) shRNAi (SMAD4) Modified Boyden chamber invasion and migration assay, electrophoretic mobility shift assay and luciferase reporter assay	SMAD4 is necessary for the upregulation of N-CAD. Knocked down SMAD4 reduces N-CAD protein levels and inhibits invasion and migration.
Whittle MC et al. [107]	2015	PDAC cell lines (CFPAC-1, PANC-1 and MiaPaCa-2) PDAC GEMM NOD SCID/NCr mice Human TMA	WB (p16, p19, p21, Parp, RUNX3 and SMAD4) IHC (amylase, cleaved caspase 3, insulin, CK19, RUNX3 and SMAD4) IF (E-CAD) RT-qPCR (Col6a1, RUNX3 and Spp1) ELISA (Spp1) shRNAi (Col6a1, RUNX3) Cell proliferation assay, migration assay, Matrigel invasion assay, soft agar assay and luciferase reporter assay	Pancreas-specific homozygous deletion of *SMAD4* in the mouse model of PDAC abrogates the TGF-beta-induced EMT of cancer epithelia but does not impair metastasis.
David JC et al. [113]	2016	Cell lines from GEMM PDAC GEMM	WB (Cdx2, cleaved caspase 3, E-CAD, Foxa2, KLF5, Pdx1, SMAD2/3, SMAD4, Snail, SOX4 and ZEB1) IHC (cleaved caspase 3, E-CAD, CK19, KLF5 and SOX4) IF (cleaved caspase 3, E-CAD, CK19, KLF5 and SOX4) ChIP (E-CAD, Klf5, Serpine, SMAD2/3 and SMAD7) shRNA screening (Foxa2, Klf5, Renill, Snail, SOX4 and ZEB1) Cleaved caspase activity measurements	The TGF-beta-/SMAD4-dependent pathway induces EMT and then apoptosis, implicating SOX4 and KLF5.
Shichi Y et al. [117]	2019	PDAC cell lines (PANC-1, MiaPaCa-2 and PK-1)	IHC (CA19.9, CEA, E-CAD, CKAE1AE3, CK7, Ki-67, phospho-SMAD2L/3L, SMAD4, TGF-beta receptor II, trypsin, and VIM) RT-qPCR (E-CAD, N-CAD, Snail and VIM) Sphere-forming assays, scanning electron microscopic analysis and transmission electron microscopic analysis	Cells with loss of SMAD4 maintain an epithelial phenotype.
Mohd Faheem M et al. [118]	2020	PDAC cell lines (PANC-1, MiaPaCa-2 and BxPC3)	WB (Akt/pAkt, Bax, Bcl2, E-CAD, CDK2, cleaved caspase 3, caspase 3, cyclin A, cyclin E, NM23H1, p21, p27, Par-4, SMAD4, Snail, STRAP and VIM) IF (E-CAD, NM23H1, Par-4, SMAD4 and Snail) RT-qPCR (Akt/pAkt, Bax, Bcl2, E-CAD, CDK2, cleaved caspase 3, caspase 3, cyclin A, cyclin E, NM23H1, p21, p27, Par-4, SMAD4, Snail, STRAP and VIM) siRNA (NM23H1, Par-4 and SMAD4) Co-immunoprecipitation assay (NM23H1, STRAP) Fluorescent gelatin degradation assay, transient transfections with N3-secAnnexinV-mVenus construct and D spheroid migration assay	Par-4 induces SMAD4 lethal EMT.
Chan-Seng-Yue M et al. [101]	2020	Tissue bulk analysis of human PDAC specimens	WGS, WTS, RNA-Seq and scRNA-Seq	Complete loss of SMAD4 is more frequent in “Classical A/B forms” and, therefore, is less associated with the upregulation of EMT markers, which are associated with “Basal A/B forms”.
EMT requires *SMAD4* alteration				
Yamada S et al. [56]	2015	Retrospective clinical trial	IHC (E-CAD, SMAD4 and VIM)	*SMAD4* inactivation was associated with tumor progression, pattern of failure and EMT. SMAD4-negative tumors correlated with a mesenchymal phenotype.
Wartenberg M et al. [121]	2018	Retrospective clinical trial	IHC (CD3, CD4, CD8, CD20, FOXP3, MLH1, MSH2, MSH6, p63, PD-L1 and PMS2) NGS (APC, ATM, CDKN2A, EGFR, FGFR3, GNAS, JAK3, KRAS, MET, PIK3CA, SMAD4, SMARCB1, STK11 and TP53)	EMT-like tumor budding and *SMAD4* inactivation are associated with the “immune-escape” phenotype.
Wang Z et al. [52]	2019	Retrospective clinical trial PDAC cell lines (PANC-1, SW1990)	WB (E-CAD, SMAD4 and VIM) IHC (SMAD4) RT-qPCR (E-CAD, SMAD4 and VIM) Sanger sequencing (SMAD4) Cell proliferation assay, transwell migration assay and wound-healing assay	The *SMAD4* Y353C mutation leads to increased cell migration as well as invasion and EMT promotion in vitro.
EMT is SMAD4 independent				
Levy L et al. [122]	2005	PDAC cell line (Colo-357) Human keratinocyte cell line (HaCaT)	WB (HA, PAI-1, p21, SMAD2/3, phosphor-SMAD2, phosphor-SMAD3, SMAD4 and Smurf1) IF (E-CAD, VIM) RT-PCR siRNA (SMAD4) Luciferase assay, cell cycle analysis and scratch assays	SMAD4 is necessary for TGF-beta-induced cell cycle arrest and migration but is not involved in the TGF-beta-induced EMT.

ChIP: chromatin immunoprecipitation; Beta-CAT: beta-catenin; E-CAD: E-cadherin; EMT: epithelial–mesenchymal transition; GEMM: genetically engineered mouse model; IF: immunofluorescence; IHC: immunohistochemistry; CKAE1AE3: cytokeratin AE1AE3; CK7: cytokeratin 7; CK19: cytokeratin 19; NB: Northern blot; N-CAD: N-cadherin; PDAC: pancreatic ductal adenocarcinoma; RNA-Seq: RNA sequencing; scRNA-Seq: single-cell RNA sequencing; RT-PCR: reverse transcription polymerase chain reaction; RT-qPCR: reverse transcription quantitative polymerase chain reaction; shRNA: short hairpin RNA; shRNAi: short hairpin RNA interference; siRNA: small interfering RNA; SMA: smooth muscle actin; TGF-beta: transforming growth factor-beta; TMA: tissue microarray; VIM: vimentin; WB: Western blot; WGS: whole-genome sequencing; WTS: whole-transcriptome sequencing.

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
