# Peer review of "The Role of SMAD4 Inactivation in Epithelial–Mesenchymal Plasticity of Pancreatic Ductal Adenocarcinoma: The Missing Link?"

_cancers, 2022, doi:10.3390/cancers14040973_

Round 1

Reviewer 1 Report

The authors present a comprehensive review of the role of Smad 4 in EMT in pancreatic cancer.

The manuscript has a clear structure, and is altogether well-written. I would suggest shortening the general section about EMT in PDAC, since this shifts the focus of the review. In addition, some repetitions e.g. those concerning the possible role of Smad4 in the metastatic process and its prognostic significance should be avoided. One or two schematic figures illustrating the TGF-pathway and the role of Smad4 in EMT should be added to improve readability.

Further minor points:

  • The authors should carefully review the reference section, since it is almost completely wrong (e.g. p. 5, paragraph 6 the text refers to the work of Jones et al. and mentions ref. 11, but the correct ref. according to the list is number 9; p. 6 ref. 17 is cited as being the work of Biankin et al, but this is ref. 10. And so on!).
  • Figure 1a is definitely overstained and should be replaced by a better one. Nuclear staining is obfuscated by the very strong cytoplasmic staining.
  • Punctuation should be revised throughout the manuscript

Reviewer 2 Report

The article entitled “The role of SMAD4 inactivation in Epithelial Mesenchymal Plasticity of Pancreatic Ductal Adenocarcinoma: The Missing Link” describes comprehensive view of role of SMAD4, EMT, and EMP. Here authors have made decent literature review to discuss SMAD4 controlled TGF-B induced EMT in PDAC. However, there are many major and minor flaws needs to be fixed before accepting the manuscript for publication.

Major Concern:

  1. The authors have discussed many published articles to submit a point, however what the writers think about the point is missing. Please discuss your views about that in the discussion section.
  2. The article has unnecessarily detailed description of processes like ‘An introduction of EMT’, EMT in PDAC, etc. Here, the reviewer think that a brief description of the topic would suffice and keep the reader engaged with the manuscript.

Minor Concerns:

  1. Please use English writing service at some of the places it requires language correction and please use scientific language.
  2. Majority of the sentences are redundant, such as SMAD4 inactivation is relatively specific to gastrointestinal cancers and occurs in about 50% of PDAC and few others. Please update this.
  3. Many sentences in manuscript are very long (4-6 lines), which loss the track of the original interest in the manuscript. It is always advisable to keep lines short (1-2 sentences are good enough).
  4. SMAD4 and Smad 4 has been used interchangeably, please update that.
  5. The abbreviation section should move from center to the beginning of the manuscript.
  6. Please update the references, such as 15, 41, 52, 89, 96, etc. and please maintain the style as per journals guideline.

Reviewer 3 Report

In this review article, the authors described the controversy in the association of SMAD4 inactivation in PDAC and TGF-beta-induced EMT or EMP, as well as the metastasis and prognosis.

It is known that TGF-beta-SMAD signal has dual roles, a tumor suppressor at the early carcinogenesis stage, and a tumor promoter at the advanced stage. At the advanced stage, the tumor promoting effect is frequently discussed with the EMT and metastasis process, however, previous studies also demonstrated that SMAD4 loss is associated with frequent metastasis and poor prognosis. The authors introduced a number of previous reports by categorizing them into SMAD4-dependent EMT, SMAD4 loss-dependent EMT, and SMAD4-independent EMT, and raised possible points of issue, the existence of EMP, not only EMT, the possible inconsistency in the evaluation of EMT (and EMP), and also the diversity of patients’ stages in the analyses of prognosis, etc.

In general, this review is well written. I raised a couple of points.

1) The authors discussed TGF-beta-induced EMT, however, there might also be EMT induced by other impulse than TGF-beta, and it might be correlated with SMAD4 inactivation and metastasis. Is there any discussion about this?

2) As for TGF-beta signaling, Smad4-null tumors showed more epithelial identity in genetically-engineered mouse models harboring Kras mutation, as cited in Table 1 (Bardeesy et al., Genes Dev 2006), which was consistent with the knockout of TGF-beta receptor II (Tgfbr2) (Ijichi et al., Genes Dev 2006), in contrast, Tp53 mutation or p16(Ink4a) loss in the same context showed sarcomatous tumors up to 30% (Hingorani et al., Cancer Cell 2005; Aguirre et al., Genes Dev 2003). This also suggested the correlation of intact TGF-beta signaling with EMT in the tumor phenotype, the authors may want to add in the discussion.

3) In Table1, the authors described, “TGF-beta-induced EMT is Smad4 loss-dependent”, however, is it really TGF-beta-induced? Especially, in the retrospective clinical trials, lots of IHC were performed, but, it is unclear whether the EMT is TGF-beta-induced or not.

4) “Smad4 loss-dependent EMT” is a somewhat surprising phenomenon. Could the authors describe underlying mechanisms a bit more according to the cell line experiments etc. from the cited references?

5) There are typos and grammatical errors. The references are wrongly numbered off by one.
